# Safe Exploration for Interactive Machine Learning

**Matteo Turchetta**
Dept. of Computer Science
ETH Zurich
matteotu@inf.ethz.ch

Felix Berkenkamp
Dept. of Computer Science
ETH Zurich
befelix@inf.ethz.ch

Andreas Krause
Dept. of Computer Science
ETH Zurich
krausea@ethz.ch

## Abstract

In Interactive Machine Learning (IML), we iteratively make decisions and obtain
noisy observations of an unknown function. While IML methods, e.g., Bayesian
optimization and active learning, have been successful in applications, on real-
world systems they must provably avoid unsafe decisions. To this end, safe IML
algorithms must carefully learn about *a priori* unknown constraints without making
unsafe decisions. Existing algorithms for this problem learn about the safety of all
decisions to ensure convergence. This is sample-inefficient, as it explores decisions
that are not relevant for the original IML objective. In this paper, we introduce a
novel framework that renders any existing unsafe IML algorithm safe. Our method
works as an add-on that takes suggested decisions as input and exploits regularity
assumptions in terms of a Gaussian process prior in order to efficiently learn about
their safety. As a result, we only explore the safe set when necessary for the
IML problem. We apply our framework to safe Bayesian optimization and to safe
exploration in deterministic Markov Decision Processes (MDP), which have been
analyzed separately before. Our method outperforms other algorithms empirically.

## 1   Introduction

Interactive Machine Learning (IML) problems, where an autonomous agent actively queries an
unknown function to optimize it, learn it, or otherwise act based on the observations made, are
pervasive in science and engineering. For example, Bayesian optimization (BO) (Mockus et al., 1978)
is an established paradigm to optimize unknown functions and has been applied to diverse tasks
such as optimizing robotic controllers (Marco et al., 2017) and hyperparameter tuning in machine
learning (Snoek et al., 2012). Similarly, Markov Decision Processes (MDPs) (Puterman, 2014) model
sequential decision making problems with long term consequences and are applied to a wide range of
problems including finance and management of water resources (White, 1993).

However, real-world applications are subject to safety constraints, which cannot be violated during
the learning process. Since the dependence of the safety constraints on the decisions is unknown *a
priori*, existing algorithms are not applicable. To *optimize* the objective without violating the safety
constraints, we must *carefully explore* the space and ensure that decisions are safe before evaluating
them. In this paper, we propose a data-efficient algorithm for safety-constrained IML problems.

**Related work**     One class of IML algorithms that consider safety are those for BO with Gaussian
Process (GP) (Rasmussen, 2004) models of the objective. While classical BO algorithms focus
on efficient optimization (Srinivas et al., 2010; Thompson, 1933; Wang and Jegelka, 2017), these
methods have been extended to incorporate safety constraints. For example, Gelbart et al. (2014)
present a variant of expected improvement with unknown constraints, while Hernández-Lobato et al.
(2016) extend an information-theoretic BO criterion to handle black-box constraints. However, these
methods only consider finding a safe solution, but allow unsafe evaluations during the optimization
process. Wu et al. (2016) define safety as a constraint on the cumulative reward, while Schreiter et al.
(2015) consider the safe exploration task on its own. The algorithms SAFEOPT (Sui et al., 2015;

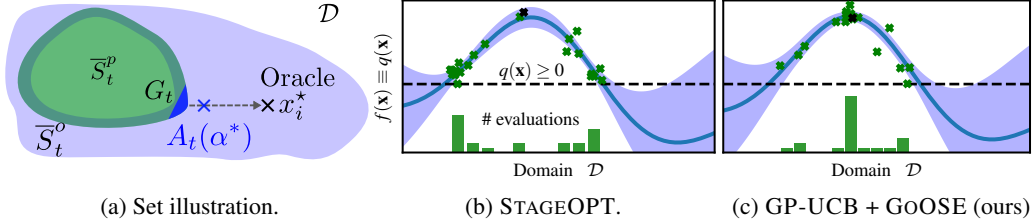

| (a) Set illustration. | (b) STAGEOPT. | (c) GP-UCB + GOOSE (ours) |

Figure 1: Existing algorithms for safe IML aim to expand the safe set $\bar{S}^p$ (green shaded) in Fig. 1a by evaluating decisions on the boundary of the pessimistic safe set (dark green shaded). This can be inefficient: to solve the safe BO problem in Fig. 1b, STAGEOPT evaluates decisions (green crosses, histogram) close to the safety constraint $q(\cdot) > 0$ (black dashed), even though the maximum (black cross) is known to be safe. In contrast, our method uses decisions $\mathbf{x}_i^\star$ from existing *unsafe* IML algorithms (oracle) within the optimistic safe set $\bar{S}_t^{o,\epsilon}$ (blue shaded, Fig. 1a). It can then use any heuristic to select learning targets $A_t$ (blue cross) that are informative about the safety of $\mathbf{x}_i^\star$ and learns about them efficiently within $G_t \subseteq \bar{S}_t^p$ (blue shaded region). Since this method only learns about the safe set when necessary, we evaluate more close-to-optimal decisions in Fig. 1c.

Berkenkamp et al., 2016) and STAGEOPT (Sui et al., 2018) both guarantee safety of the exploration and near-optimality of the solution. However, they treat the exploration of the safe set as a proxy objective, which leads to sample-inefficient exploration as they explore the entire safe set, even if this is not necessary for the optimization task, see the evaluation counts (green) in Fig. 1b for an example.

Safety has also been investigated in IML problems in directed graphs, where decisions have long-term effects in terms of safety. Moldovan and Abbeel (2012) address this problem in the context of discrete MDPs by optimizing over ergodic policies, i.e., policies that are able to return to a known set of safe states with high probability. However, they do not provide exploration guarantees. Biyik et al. (2019) study the ergodic exploration problem in discrete and deterministic MDPs with unknown dynamics and noiseless observations. Turchetta et al. (2016) investigate the ergodic exploration problem subject to unknown external safety constraints under the assumption of known dynamics by imposing additional ergodicity constraints on the SAFEOPT algorithm. Wachi et al. (2018) compute approximately optimal policies in the same context but do not actively learn about the constraint. In continuous domains, safety has been investigated by, for example, Akametalu et al. (2014); Koller et al. (2018). While these methods provide safety guarantees, current exploration guarantees rely on uncertainty sampling on a discretized domain (Berkenkamp et al., 2017). Thus, their analysis can benefit from the more efficient, goal-oriented exploration introduced in this paper.

**Contribution**    In this paper, we introduce the Goal Oriented Safe Exploration algorithm, GOOSE; a novel framework that works as an add-on to existing IML algorithms and renders them safe. Given a possibly unsafe suggestion by an IML algorithm, it safely and efficiently learns about the safety of this decision by exploiting continuity properties of the constraints in terms of a GP prior. Thus, unlike previous work, GOOSE only learns about the safety of decisions relevant for the IML problem. We analyze our algorithm and prove that, with high probability, it only takes safe actions while learning about the safety of the suggested decisions. On safe BO problems, our algorithm leads to a bound on a natural notion of safe cumulative regret when combined with a no-regret BO algorithm. Similarly, we use our algorithm for the safe exploration in deterministic MDPs. Our experiments show that GOOSE is significantly more data-efficient than existing methods in both settings.

## 2   Problem Statement and Background

In IML, an agent iteratively makes decisions and observes their consequences, which it can use to make better decisions over time. Formally, at iteration $i$, the agent $\mathcal{O}_i$ uses the previous $i-1$ observations to make a new decision $\mathbf{x}_i^\star = \mathcal{O}_i(\mathcal{D}_i)$ from a finite decision space $\mathcal{D}_i \subseteq \mathcal{D} \subseteq \mathbb{R}^d$. It then observes a noisy measurement of the unknown objective function $f : \mathcal{D} \to \mathbb{R}$ and uses the new information in the next iteration. This is illustrated in the top-left corner (blue shaded) of Fig. 2. Depending on the goal of the agent, this formulation captures a broad class of problems and many solutions to these problems have been proposed. For example, in *Bayesian optimization* the agent aims to find the global optimum $\max_{\mathbf{x}} f(\mathbf{x})$ (Mockus et al., 1978). Similarly, in active learning

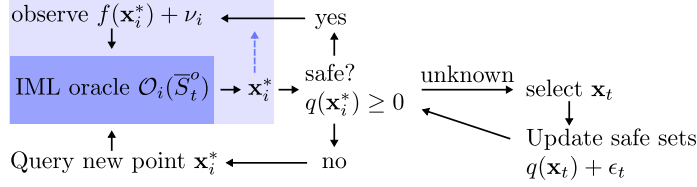

Figure 2: Overview of GoOSE. If the oracle's suggestion $\mathbf{x}_i^\star$ is safe, it can be evaluated. This is equivalent to the standard *unsafe* IML pipeline (top-left, blue shaded) in Fig. 2. Otherwise, GoOSE learns about the safety of $\mathbf{x}_i^\star$ by actively querying observations at decisions $\mathbf{x}_t$. Any provably unsafe decision is removed from the decision space and we query a new $\mathbf{x}_i^\star$ without providing a new observation of $f(\mathbf{x}_i^\star)$.

(Schreiter et al., 2015), one aims to learn about the function $f$. In the general case, the decision process may be stateful, e.g., as in dynamical systems, so that the decisions $\mathcal{D}_i$ available to the agent depend on those made in the past. This dependency among decisions can be modeled with a *directed graph*, where nodes represent decisions and an edge connects node $\mathbf{x}$ to node $\mathbf{x}'$ if the agent is allowed to evaluate $\mathbf{x}'$ given that it evaluated $\mathbf{x}$ at the previous decision step. In the BO setting, the graph is fully-connected and any decision may be evaluated, while in a deterministic MDP decisions are states and edges represent transitions (Turchetta et al., 2016).

In this paper, we consider IML problems with safety constraints, which frequently occur in real-world settings. The safety constraint can be written as $q(\mathbf{x}) \geq 0$ for some function $q$. Any decision $\mathbf{x}_i^\star$ for $i \geq 1$ evaluated by the agent must be safe. For example, Berkenkamp et al. (2016) optimize the control policy of a flying robot and must evaluate only policies that induce trajectories satisfying given constraints. However, it is unknown *a priori* which policy parameters induce safe trajectories. Thus, we do not know which decisions are safe in advance, that is, $q : \mathcal{D} \to \mathbb{R}$ is *a priori* unknown. However, we can learn about the safety constraint by selecting decisions $\mathbf{x}_t$ and obtaining noisy observations of $q(\mathbf{x}_t)$. We denote queries to $f$ with $\mathbf{x}_i^\star$ and queries to $q$ with $\mathbf{x}_t$. As a result, we face a two-tiered *safe exploration* problem: On one hand we have to safely learn about the constraint $q$ to determine which decisions are safe, while on the other hand we want to learn about $f$ to solve the IML problem. The goal is to minimize the number of queries $\mathbf{x}_t$ required to solve the IML problem.

**Regularity**     Without further assumptions, it is impossible to evaluate decisions without violating the safety constraint $q$ (Sui et al., 2015). For example, without an initial set of decisions that is known to be safe *a priori*, we may fail at the first step. Moreover, if the constraint does not exhibit any regularity, we cannot infer the safety of decisions without evaluating them first. We assume that a small initial safe set of decisions, $S_0$, is available, which may come from domain knowledge. Additionally, we assume that $\mathcal{D}$ is endowed with a positive definite kernel function, $k(\cdot, \cdot)$, and that the safety constraint $q$ has bounded norm in the induced *Reproducing Kernel Hilbert Space* (RKHS) (Schölkopf and Smola, 2002)), $\|q\|_k \leq B_q$. The RKHS norm measures the smoothness of the safety feature with respect to the kernel, so $q$ is $L$-Lipschitz continuous with respect to the kernel metric $d(\mathbf{x}, \mathbf{x}') = \sqrt{k(\mathbf{x}, \mathbf{x}) - 2k(\mathbf{x}, \mathbf{x}') + k(\mathbf{x}', \mathbf{x}')}$ with $L = B_q$ (Steinwart and Christmann, 2008, (4.21))

This assumption allows us to model the safety constraint function $q$ with a GP (Rasmussen, 2004). A GP is a distribution over functions parameterized by a mean function $\mu(\cdot)$ and a covariance function $k(\cdot, \cdot)$. We set $\mu(\mathbf{x}) = 0$ for all $\mathbf{x} \in \mathcal{D}$ without loss of generality. The covariance function encodes our assumptions about the safety constraint. Given $t$ observations of the constraint $\mathbf{y} = (q(\mathbf{x}_1) + \eta_1, \ldots, q(\mathbf{x}_t) + \eta_t)$ at decisions $\mathcal{D}_t = \{\mathbf{x}_n\}_{n=1}^t$, where $\eta_n \sim \mathcal{N}(0, \sigma^2)$ is a zero-mean i.i.d. Gaussian noise, the posterior belief is distributed as a GP with mean, covariance, and variance

$$\mu_t(\mathbf{x}) = \mathbf{k}_t^{\mathrm{T}}(\mathbf{x})(\mathbf{K}_t + \sigma^2 \mathbf{I})^{-1} \mathbf{y}_t, \; k_t(\mathbf{x}, \mathbf{x}') = k(\mathbf{x}, \mathbf{x}') - \mathbf{k}_t^{\mathrm{T}}(\mathbf{x})(\mathbf{K}_t + \sigma^2 \mathbf{I})^{-1} \mathbf{k}_t(\mathbf{x}'), \; \sigma_t(\mathbf{x}) = k_t(\mathbf{x}, \mathbf{x})$$

respectively. Here, $\mathbf{k}_t(\mathbf{x}) = (k(\mathbf{x}_1, \mathbf{x}), \ldots, k(\mathbf{x}_t, \mathbf{x}))$, $\mathbf{K}_t$ is the positive definite kernel matrix $[k(\mathbf{x}, \mathbf{x}')]_{\mathbf{x}, \mathbf{x}' \in D_t}$, and $\mathbf{I} \in \mathbb{R}^{t \times t}$ denotes the identity matrix.

**Safe decisions**     The previous regularity assumptions can be used to determine which decisions are safe to evaluate. Our classification of the decision space is related to the one by Turchetta et al. (2016), which combines non-increasing and reliable confidence intervals on $q$ with a reachability analysis of the underlying graph structure for decisions. Based on a result by Chowdhury and Gopalan (2017), they use the posterior GP distribution to construct confidence bounds

$l_t(\mathbf{x}) := \max(l_{t-1}(\mathbf{x}), \mu_{t-1}(\mathbf{x}) - \beta_t\sigma_{t-1}(\mathbf{x}))$ and $u_t(\mathbf{x}) := \min(u_{t-1}(\mathbf{x}), \mu_{t-1}(\mathbf{x}) + \beta_t\sigma_{t-1}(\mathbf{x}))$ on the function $q$. In particular, we have $l_t(\mathbf{x}) \le q(\mathbf{x}) \le u_t(\mathbf{x})$ with high probability when the scaling factor $\beta_t$ is chosen as in Theorem 1. Thus, any decision $\mathbf{x}$ with $l_t(\mathbf{x}) \ge 0$ is satisfies the safety constraint $q(\mathbf{x}) \ge 0$ with high probability.

To analyze the exploration behavior of their algorithm, Turchetta et al. (2016) use the confidence intervals within the current safe set, starting from $S_0$, and the Lipschitz continuity of $q$ to define $S_t^p$, the set of decisions that satisfy the constraint with high probability. We use a similar, albeit more efficient, definition in Sec. 3. In practice, one may use the confidence intervals directly. Moreover, in order to avoid exploring decisions that are instantaneously safe but that would force the agent to eventually evaluate unsafe ones due to the graph structure $\mathcal{G}$, Turchetta et al. (2016) define $\bar{S}_t^p$, the subset of safe and ergodic decisions, i.e., decisions that are safe to evaluate in the short and long term.

**Previous Exploration Schemes**   Given that only decisions in $\bar{S}_t^p$ are safe to evaluate, any *safe* IML algorithm faces an extended exploration-exploitation problem: it can either optimize decisions within $\bar{S}_t^p$, or expand the set of safe decisions in $\bar{S}_t^p$ by evaluating decisions on its boundary. Existing solutions to the safe exploration problem in both discrete and continuous domains either do not provide theoretical exploration guarantees (Wachi et al., 2018) or treat the exploration of the safe set as a proxy objective for optimality. That is, the methods uniformly reduce uncertainty on the boundary of the safe set in Fig. 1a until the entire safe set is learned. Since learning about the entire safe set is often unnecessary for the IML algorithm, this procedure can be sample-inefficient. For example, in the safe BO problem in Fig. 1b with $f = q$, this exploration scheme leads to a large number of unnecessary evaluations on the boundary of the safe set.

## 3  Goal-oriented Safe Exploration (GOOSE)

In this section, we present our algorithm, GOOSE. We do not propose a new *safe* algorithm for a specific IML setting, but instead exploit that, for specific *IML* problems high-performance, *unsafe* algorithms already exist. We treat any such *unsafe* algorithm as an IML oracle $\mathcal{O}_i(S)$, which, given a domain $S$ and $i - 1$ observations of $f$, suggests a new decision $\mathbf{x}_i^\star \in S$, see Fig. 2 (blue shaded).

GOOSE can extend any such *unsafe* IML algorithm to the safety-constrained setting. Thus, we effectively leave the problem of querying $f$ to the oracle and only consider safety. Given an *unsafe* oracle decision $\mathbf{x}_i^\star$, GOOSE only evaluates $f(\mathbf{x}_i^\star)$ if the decisions $\mathbf{x}_i^\star$ is known to be safe. Otherwise, it *safely learns* about $q(\mathbf{x}_i^\star)$ by safely and efficiently collecting observations $q(\mathbf{x}_t)$. Eventually it either learns that the decision $\mathbf{x}_i^\star$ is safe and allows the oracle to evaluate $f(\mathbf{x}_i^\star)$, or that $\mathbf{x}_i^\star$ cannot be guaranteed to be safe given an $\epsilon$-accurate knowledge of the constraint, in which case the decision set of the oracle is restricted and a new decision is queried, see Fig. 2.

Previous approaches treat the expansion of the safe set as a proxy-objective to provide completeness guarantees. Instead, GOOSE employs goal-directed exploration scheme with a novel theoretical analysis that shifts the focus from greedily reducing the uncertainty *inside* the safe set to learning about the safety of decisions *outside* of it. This scheme retains the worst-case guarantees of existing methods, but is significantly more sample-efficient in practice. Moreover, GOOSE encompasses existing methods for this problem. We now describe the detailed steps of GOOSE in Alg. 1 and 2.

**Pessimistic and optimistic expansion.**   To effectively shift the focus from inside the safe set to outside of it, GOOSE must reason not only about the decisions that are currently known to be safe but also about those that could eventually be classified as safe in the future. In particular, it maintains two sets, which are an inner/outer approximation of the set of safe decisions that are reachable from $S_0$ and are based on a pessimistic/optimistic estimate of the constraint given the data, respectively.

The pessimistic safe set contains the decisions that are safe with high probability and is necessary to guarantee safe exploration (Turchetta et al., 2016). It is defined in two steps: discarding the decisions that are not instantaneously safe and discarding those that we cannot reach/return from safely (see Fig. 3b) and, thus, are not safe in the long term. To characterize it starting from a given set of safe decisions $S$, we define the pessimistic constraint satisfaction operator,

$$p_t(S) = \{\mathbf{x} \in \mathcal{D}, \,|\, \exists \mathbf{z} \in S : l_t(\mathbf{z}) - Ld(\mathbf{x}, \mathbf{z}) \ge 0\}, \tag{1}$$

which uses the lower bound on the safety constraint of the decisions in $S$ and the Lipschitz continuity of $q$ to determine the decisions that instantaneously satisfy the constraint with high probability, see Fig. 3a. However, for a general graph $\mathcal{G}$, decisions in $p_t(S)$ may be unsafe in the long-term

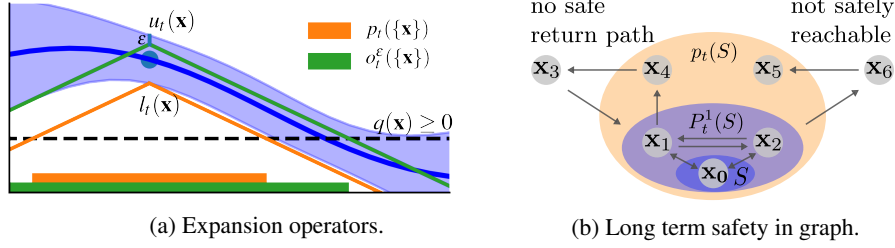

| (a) Expansion operators. | (b) Long term safety in graph. |

Figure 3: Fig. 3a shows the pessimistic and optimistic constraint satisfaction operators that use the confidence intervals on the constraint and its Lipschitz continuity to make inference about the safety of decisions that have not yet been evaluated. Fig. 3b illustrates the long-term safety definition. While decisions in $p_t(S)$ are myopically safe, decisions in $P_t^1(S)$ are safe in the long-term. This excludes $\mathbf{x}_4$ and $\mathbf{x}_5$, as no safe path from/to them exists.

as in Fig. 3b: No safe path to the decision $\mathbf{x}_5$ exists, so that it can not be safely *reached*. Similarly, if we were to evaluate $\mathbf{x}_4$, the graph structure forces us to eventually evaluate $\mathbf{x}_3$, which is not contained in $p_t(S)$ and might be unsafe. That is, we cannot safely *return* from $\mathbf{x}_4$. To exclude these decisions, we use the ergodicity operator introduced by Turchetta et al. (2016), which allows us to find those decisions that are pessimistically safe in the short and long term $P_t^1(S) = p_t(S) \cap R^{\mathrm{ergodic}}(p_t(S), S)$ (see Appendix A or (Turchetta et al., 2016) for the definition of $R^{\mathrm{ergodic}}$). Alternating these operations $n$ times, we obtain the $n$-step pessimistic expansion operator, $P_t^n(S) = p_t(P_t^{n-1}(S)) \cap R^{\mathrm{ergodic}}(p_t(P_t^{n-1}(S)), S)$, which, after a finite number of steps, converges to its limiting set $\tilde{P}_t(S) = \lim_{n \to \infty} P_t^n(S)$.

The optimistic safe set excludes the decisions that are unsafe with high probability and makes the exploration efficient by restricting the decision space of the oracle. Similarly to the pessimistic one, it is defined in two steps. However, it uses the following optimistic constraint satisfaction operator,

$$o_t^\epsilon(S) = \{\mathbf{x} \in \mathcal{D}, \, | \, \exists \mathbf{z} \in S : u_t(\mathbf{z}) - Ld(\mathbf{x}, \mathbf{z}) - \epsilon \geq 0\}. \tag{2}$$

See Fig. 3a for a graphical intuition. The additional $\epsilon$-uncertainty term in the optimistic operator accounts for the fact that we only have access to noisy measurements of the constraint and, therefore, we can only learn it up to a specified statistical accuracy. The definitions of the optimistic expansion operators $O_t^{\epsilon,n}(S)$ and $\tilde{O}_t^\epsilon(S)$ are analogous to the pessimistic case by substituting $p_t$ with $o_t^\epsilon$. The sets $\tilde{P}_t$ and $\tilde{O}_t^\epsilon$ indicate the largest set of decisions that can be classified as safe in the short and long term assuming the constraint attains the worst/best possible value within $S$, given the observations available and, for the optimistic case, despite an $\epsilon$ uncertainty.

**Optimistic oracle** The IML oracle $\mathcal{O}_i(S)$ suggests decisions $\mathbf{x}_i^\star \in S$ to evaluate within a given subset $S$ of $\mathcal{D}$. To make the oracle efficient, we restrict its decision space to decisions that could optimistically be safe in the long and short term. In particular, we define the optimistic safe set $\bar{S}^{o,\epsilon}$ in Line 8 of Alg. 1 based on the optimistic expansion operator introduced above. The oracle uses this set to suggest a potentially unsafe, candidate decision $\mathbf{x}_i^\star = \mathcal{O}_i(\bar{S}_t^{o,\epsilon})$ in Line 4.

**Safe evaluation** We determine safety of the suggestion $\mathbf{x}_i^\star$ similarly to Turchetta et al. (2016) by constructing the set $\bar{S}_t^p$ of decisions that are safe to evaluate. However, while Turchetta et al. (2016) use the one step pessimistic expansion operator in their definition, $P_t^1$, we use its limit set in Line 7 of Alg. 1, $\tilde{P}_t$. While both operators eventually identify the same safe set, our definition allows for a more efficient expansion. For example, consider the case where the graph over the decision space $\mathcal{G}$ is a chain of length $m$ and where, for all $j = 1, \cdots, m$, the lower bound on the safety of decision $j - 1$ guarantees the safety of decision $j$ with high probability. In this case, Turchetta et al. (2016) require $m - 1$ iterations to fully expand the safe set, while our classification requires only one.

If we know that $\mathbf{x}_i^\star$ is safe to evaluate, i.e., $\mathbf{x}_i^\star \in \bar{S}_t^p$, then the oracle obtains a noisy observation of $f(\mathbf{x}_i^\star)$ in Line 10. Otherwise GOOSE proceeds to safely learn about the safety of $\mathbf{x}_i^\star$ using a safe expansion strategy in lines Lines 5–8 that we outline in the following. This routine is repeated until we can either include $\mathbf{x}_i^\star$ in $\bar{S}_t^p$, in which case we can safely evaluate $f(\mathbf{x}_i^\star)$, or remove it from the decision space $\bar{S}_t^{o,\epsilon}$ and query the oracle for a new suggestion.

| **Algorithm 1** GoOSE | **Algorithm 2** Safe Expansion (SE) |
|---|---|
| 1: **Inputs:** Lipschitz constant $L$, Seed $S_0$, Graph $\mathcal{G}$, Oracle $O$, Accuracy $\epsilon$. | 1: **Inputs:** $\bar{S}_t^{o,\epsilon}, \bar{S}_t^p, \mathcal{G}, \mathbf{x}^\star$ |
| 2: $\bar{S}_0^p \leftarrow S_0, \bar{S}_0^{o,\epsilon} \leftarrow \mathcal{D}, t \leftarrow 0,$ $l_0(\mathbf{x}) \leftarrow 0$ for $\mathbf{x} \in S_0$ | 2: $W_t^\epsilon \leftarrow \{\mathbf{x} \in \bar{S}_t^p \mid u_t(\mathbf{x}) - l_t(\mathbf{x}) > \epsilon\}$ |
| 3: **for** $k = 1, 2, \dots$ **do** | 3: $A_t(p) \leftarrow \{\mathbf{x} \in \bar{S}_t^{o,\epsilon} \setminus p_t^0(\bar{S}_t^p) \mid h(\mathbf{x}) = p\}$ |
| 4:    $\mathbf{x}_i^\star \leftarrow \mathcal{O}(\bar{S}_t^{o,\epsilon})$ | 4: // Highest priority targets in $A_t$ with expanders |
| 5:    **while** $\mathbf{x}_i^\star \notin \bar{S}_t^p$ **do** |     $\alpha^* \leftarrow \max \alpha$  s.t.  $\left|G_t^\epsilon(\alpha)\right| > 0$ |
| 6:       SE($\bar{S}_t^{o,\epsilon}, \bar{S}_t^p, \mathcal{G}, \mathbf{x}_i^\star$), $t \leftarrow t + 1$ | 5: **if** optimization problem feasible **then** |
| 7:       $\bar{S}_t^p \leftarrow \tilde{P}_t(\bar{S}_{t-1}^p)$ | 6:    $\mathbf{x}_t \leftarrow \text{argmax}_{\mathbf{x} \in G_t^\epsilon(\alpha^*)} w_t(\mathbf{x})$ |
| 8:       $\bar{S}_t^{o,\epsilon} \leftarrow \tilde{O}_t^\epsilon(\bar{S}_{t-1}^p)$ | 7:    Update GP with $y_t = q(\mathbf{x}_t) + \eta_t$ |
| 9:       **if** $\mathbf{x}_i^\star \notin \bar{S}_t^{o,\epsilon}$ **then go to** Line 4 | |
| 10:    Evaluate $f(\mathbf{x}_i^\star)$ and update oracle | |

**Safe expansion**    If the oracle suggestion $\mathbf{x}_i^\star$ is not considered safe, $\mathbf{x}_i^\star \notin \bar{S}_t^p$, GoOSE employs a goal-directed scheme to evaluate a safe decision $\mathbf{x}_t \in \bar{S}_t^p$ that is informative about $q(\mathbf{x}_i^\star)$, see Fig. 1a. In practice, it is desirable to avoid learning about decisions beyond a certain accuracy $\epsilon$, as the number of observations required to reduce the uncertainty grows exponentially with $\epsilon$ (Sui et al., 2018). Thus, we only learn about decisions in $\bar{S}_t^p$ whose safety values are not known $\epsilon$-accurately yet in Line 2, $W_t^\epsilon = \{\mathbf{x} \in \bar{S}_t^p \mid u_t(\mathbf{x}) - l_t(\mathbf{x}) > \epsilon\}$, where $u_t(\mathbf{x}) - l_t(\mathbf{x})$ is the width of the confidence interval at $\mathbf{x}$.

To decide which decision in $W_t^\epsilon$ to learn about, we first determine a set of learning targets outside the safe set (dark blue cross in Fig. 1a), and then learn about them efficiently within $\bar{S}_t^p$. To quantify how useful a learning target $\mathbf{x}$ is to learn about $q(\mathbf{x}_i^\star)$, we use any given iteration-dependent heuristic $h_t(\mathbf{x})$. We discuss particular choices later, but a large priority $h(\mathbf{x})$ indicates a relevant learning target (dashed line, Fig. 1a). Since $p_t^0(\bar{S}_t^p)$ denotes the decisions that are known to satisfy the constraint with high probability and $\bar{S}_t^{o,\epsilon}$ excludes the decisions that are unsafe with high probability, $\bar{S}_t^{o,\epsilon} \setminus p_t^0(\bar{S}_t^p)$ indicates the decisions whose safety we are uncertain about. We sort them according to their priority and let $A_t(\alpha)$ denote the subset of decision with equal priority.

Ideally, we want to learn about the decisions with the highest priority. However, this may not be immediately possible by evaluating decisions within $W_t^\epsilon$. Thus, we must identify the decisions with the highest priority that we can learn about starting from $W_t^\epsilon$. Therefore, similarly to the definition of the optimistic safe set, we identify decisions $\mathbf{x}$ in $W_t^\epsilon$ that have a large enough plausible value $q(\mathbf{x})$ that they could guarantee that $q(\mathbf{z}) \geq 0$ for some $\mathbf{z}$ in $A_t(\alpha)$. However, in this case, we are only interested in decisions that can be instantly classified as safe (rather than eventually). Therefore, we focus on this set of *potential immediate expanders*, $G_t^\epsilon(\alpha) = \{\mathbf{x} \in W_t^\epsilon, \mid \exists \mathbf{z} \in A_t(\alpha) \colon u_t(\mathbf{x}) - Ld(\mathbf{x}, \mathbf{z}) \geq 0\}$. In Line 4 of Alg. 2 we select the decisions with the priority level $\alpha^*$ such that there exist uncertain, safe decisions in $W_t^\epsilon$ that could allow us to classify a decision in $A_t(\alpha^*)$ as safe and thereby expand the current safe set $\bar{S}_t^p$. Intuitively, we look for the highest priority targets that can potentially be classified as safe by safely evaluating decisions that we have not already learned about to $\epsilon$-accuracy.

Given these learning targets $A_t(\alpha^*)$ (blue cross, Fig. 1a), we evaluate the most uncertain decision in $G_t^\epsilon(\alpha^*)$ (blue shaded, Fig. 1a) in Line 6 and update the GP model with the corresponding observation of $q(\mathbf{x}_t)$ in Line 7. This uncertainty sampling is restricted to a small set of decisions close to the goal. This is different from methods without a heuristic that select the most uncertain secision on the boundary of $\bar{S}^p$ (green shaded in Fig. 1a). In fact, our method is equivalent to the one by Turchetta et al. (2016) when an uninformative heuristic $h(\mathbf{x}) = 1$ is used for all $\mathbf{x}$. We iteratively select and evaluate decisions $\mathbf{x}_t$ until we either determine that $\mathbf{x}_i^\star$ is safe, in which case it is added to $\bar{S}^p$, or we prove that we can not safely learn about it for given accuracy $\epsilon$, in which case is removed from $\bar{S}^{o,\epsilon}$ and a the oracle is queried with an updated decision space for a new suggestion.

To analyze our algorithm, we define the largest set that we can learn about as $\tilde{R}_\epsilon(S_0)$. This set contains all the decisions that we could certify as safe if we used a full-exploration scheme that learns the safety constraint $q$ up to $\epsilon$ accuracy for all decisions inside the current safe set. This is a natural exploration target for our safe exploration problem (see Appendix A for a formal definition). We have the following main result, which holds for any heuristic:

**Theorem 1.** *Assume that $q(\cdot)$ is L-Lipschitz continuous w.r.t. $d(\cdot, \cdot)$ with $\|q\|_k \leq B_q$, $\sigma$-sub-Gaussian noise, $S_0 \neq \emptyset$, $q(\boldsymbol{x}) \geq 0$ for all $\boldsymbol{x} \in S_0$, and that, for any two decisions $\boldsymbol{x}, \boldsymbol{x}' \in S_0$, there is a path in the graph $\mathcal{G}$ connecting them within $S_0$. Let $\beta_t^{1/2} = B_q + 4\sigma\sqrt{\gamma_t + 1 + \ln(1/\delta)}$, then, for any $h_t : \mathcal{D} \to \mathbb{R}$, with probability at least $1 - \delta$, we have $q(\boldsymbol{x}) \geq 0$ for any $\boldsymbol{x}$ visited by GOOSE. Moreover, let $\gamma_t$ denote the information capacity associated with the kernel $k$ and let $t^*$ be the smallest integer such that $\frac{t^*}{\beta_{t^*} \gamma_{t^*}} \geq \frac{C |\tilde{R}_0(S_0)|}{\epsilon^2}$, with $C = 8/\log(1 + \sigma^{-2})$, then there exists a $t \leq t^*$ such that, with probability at least $1 - \delta$, $\tilde{R}_\epsilon(S_0) \subseteq \bar{S}_t^{o,\epsilon} \subseteq \bar{S}_t^p \subseteq \tilde{R}_0(S_0)$.*

Theorem 1 guarantees that GOOSE is safe with high probability. Moreover, for any priority function $h$ in Alg. 2, it upper bounds the number of measurements that Alg. 1 requires to explore the largest safely reachable region $\tilde{R}_\epsilon(S_0)$. Note that GOOSE only achieves this upper bound if it is required by the IML oracle. In particular, the following is a direct consequence of Theorem 1:

**Corollary 1.** *Under the assumptions of Theorem 1, let the IML oracle be deterministic given the observations. Then there exists a set $S$ with $\tilde{R}_\epsilon(S_0) \subseteq S \subseteq \tilde{R}_0(S_0)$ so that $\boldsymbol{x}_i^\star = \mathcal{O}(S)$ for all $k \geq 1$.*

That is, the oracle decisions $\mathbf{x}_i^\star$ that we end up evaluating are the same as those by an oracle that was given the safe set $S$ in Corollary 1 from the beginning. This is true since the set $\bar{S}_t^{o,\epsilon}$ converges to this set $S$. Since Theorem 1 bounds the number of safety evaluations by $t^*$, Corollary 1 implies that, up to $t^*$ safety evaluations, GOOSE retains the properties (e.g., no-regret) of the IML oracle $\mathcal{O}$ over $S$.

**Choice of heuristic** While our worst-case guarantees hold for *any* heuristic, the empirical performance of GOOSE depends on this choice. We propose to use the graph structure directly and additionally define a positive cost for each edge between two nodes. For a given edge cost, we define $c(\mathbf{x}, \mathbf{x}_k^\star, \bar{S}_t^{o,\epsilon})$ as the cost of the minimum-cost path from $\mathbf{x}$ to $\mathbf{x}_k^\star$ within the optimistic safe set $\bar{S}_t^{o,\epsilon}$, which is equal to $\infty$ if a path does not exist, and we consider the priority $h(\mathbf{x}) = -c(\mathbf{x}, \mathbf{x}_k^\star, \bar{S}_t^{o,\epsilon})$. Thus, the node $\mathbf{x}$ with the lowest-cost path to $\mathbf{x}_k^\star$ has the highest priority. This reduces the design of a general heuristic to a more intuitive weight assignment problem, where the edge costs determine the planned path for learning about $\mathbf{x}_k^\star$ (dashed line in Fig. 1a). One option for the edge cost is the inverse mutual information between $\mathbf{x}$ and the suggestion $\mathbf{x}_k^\star$, so that the resulting paths contain nodes that are informative about $\mathbf{x}_k^\star$. Alternatively, having successive nodes in the path close to each other under the metric $d(\cdot, \cdot)$, so that they can be easily added to the safe set and eventually lead us to $\mathbf{x}_k^\star$, can be desirable. Thus, increasing monotone functions of the metric $d(\cdot, \cdot)$ can be effective edge costs.

# 4 Applications and Experiments

In this section, we introduce two safety-critical IML applications, discuss the consequences of Theorem 1 for these problems, and empirically compare GOOSE to stae-of-the-art competing methods. In our experiments, we set $\beta_t = 3$ for all $t \geq 1$ as suggested by Turchetta et al. (2016). This choice of $\beta_t$ ensures safety in practice, but leads to more efficient exploration than the theoretical choice in Theorem 1 (Turchetta et al., 2016; Wachi et al., 2018). Moreover, since in practice it is hard to estimate the Lipschitz constant of an unknown function, in our experiments we use the confidence intervals to define the safe set and the expanders as suggested by Berkenkamp et al. (2016).

## 4.1 Safe Bayesian optimization

In safe BO we want to optimize the unknown function $f$ subject to the unknown safety constraint $q$, see Sec. 2. In this setting, we aim to find the best input over the largest set we can hope to explore safely, $\tilde{R}_\epsilon(S_0)$. The performance of an agent is measured in terms of the $\epsilon$-safe regret $\operatorname{argmax}_{\mathbf{x} \in \tilde{R}_\epsilon(S_0)} f(\mathbf{x}) - f(\mathbf{x}_t)$ of not having evaluated the function at the optimum in $\tilde{R}_\epsilon(S_0)$.

We combine GOOSE with the unsafe GP-UCB (Srinivas et al., 2010) algorithm as an oracle. For computational efficiency, we do not use a fully connected graph, but instead connect decisions only to their immediate neighbors as measured by the kernel and assign equal weight to each edge for the heuristic $h$. We compare GOOSE to SAFEOPT (Sui et al., 2015) and STAGEOPT (Sui et al., 2018) in terms of $\epsilon$-*safe average regret*. Both algorithms use safe exploration as a proxy objective, see Fig. 1.

We optimize samples from a GP with zero mean and Radial Basis Function (RBF) kernel with variance 1.0 and lengthscale 0.1 and 0.4 for a one-dimensional and two-dimensional, respectively.

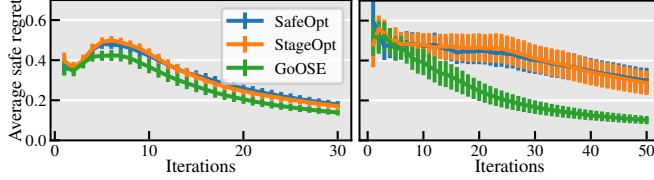

Table 1: Mars experiment performance normalized to SMDP in terms of samples to find the first path, exploration cost and computation time per iteration.

Figure 4: Average normalized $\epsilon$-safe regret for the safe optimization of GP samples over 40 (d=1, left) and 10 (d=2, right) samples. GOOSE only evaluates inputs that are relevant for the BO problem and, therefore, it converges faster than its competitors.

| | GOOSE | SEO |
|---|---|---|
| Sample | **30.0** % | 38.4 % |
| Cost | 12.7 % | **0.7** % |
| Time | **37.8** % | 518 % |

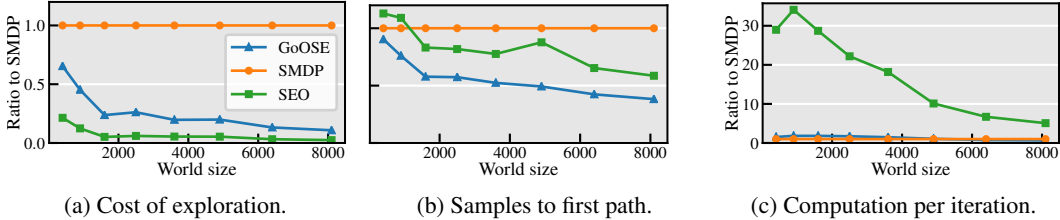

(a) Cost of exploration.  (b) Samples to first path.  (c) Computation per iteration.

Figure 5: Performance of GOOSE and SEO normalized to SMDP in terms of exploration cost, samples to find the first path and computation time per iteration as a function of the world size.

The observations are perturbed by i.i.d Gaussian noise with $\sigma = 0.01$. For simplicity, we set the objective and the constraint to be the same, $f = q$. Fig. 4 (left) shows the average regret as a function of the number of evaluations $k + t$ averaged over 40 different samples from the GP described above over a one dimensional domain (200 points evenly distributed in $[-1, 1]$). Fig. 4 (right) shows similar results averaged over 10 samples for a two dimensional domain ($25 \times 25$ uniform grid in $[0, 1]^2$).

These results confirm the intuition from Fig. 1 that using safe exploration as a proxy objective reduces the empirical performance of safe BO algorithms. The impact is more evident in the two dimensional case where there are more points along the boundaries that are nor relevant to the optimization and that are evaluated for exploration purposes.

## 4.2 Safe shortest path in deterministic MDPs

The graph that we introduced in Sec. 2 can model states (nodes) and state transitions (edges) in deterministic, discrete MDPs. Hence, GOOSE naturally extends to the goal-oriented safe exploration problem in these models. We aim to find the minimum-cost safe path from a starting state $\mathbf{x}^\dagger$ to a goal state $\mathbf{x}^\star$, without violating the unknown safety constraint, $q$. At best, we can hope to find the path within the largest safely learnable set $\tilde{R}_\epsilon(S_0)$ as in Theorem 1 with cost $c(\mathbf{x}^\dagger, \mathbf{x}^\star, \tilde{R}_\epsilon(S_0))$.

**Algorithms** We compare GOOSE to SEO (Wachi et al., 2018) and SMPD (Turchetta et al., 2016) in terms of samples required to discover the first path, total exploration cost and computation cost on synthetic and real world data. The SMDP algorithm cannot take goals into account and serves as a benchmark for comparison. The SEO algorithm aims to safely learn a near-optimal policy for any given cost function and can be adapted to the safe shortest path problem by setting the cost to $c(\mathbf{x}) = -\|\mathbf{x} - \mathbf{x}^\star\|_1$. However, it cannot guarantee that a path to $\mathbf{x}^\star$ is found, if one exists. Since the goal $\mathbf{x}^\star$ is fixed, GOOSE does not need an oracle. For the heuristic we use and optimistic estimate of the cost of the safe shortest path from $\mathbf{x}^\dagger$ to $\mathbf{x}^\star$ passing through $\mathbf{x}$; that is $h_t(\mathbf{x}) = -\min_{\mathbf{x}' \in Pred(\mathbf{x})} c(\mathbf{x}^\dagger, \mathbf{x}, \bar{S}_t^p) + \kappa c(\mathbf{x}, \mathbf{x}^\star, \bar{S}_t^{o,\epsilon})$. The first term is a conservative estimate of the safe optimal cost from $\mathbf{x}^\dagger$ to the best predecessor of $\mathbf{x}$ in $\mathcal{G}$ and the second term is an optimistic estimate of the safe optimal cost from $\mathbf{x}$ to $\mathbf{x}^\star$ multiplied by $\kappa > 1$ to encourage early path discovery. Here, we use the predecessor node because $\phi_t(\mathbf{x}) = \infty$ for all $\mathbf{x}$ not in $\bar{S}_t^p$. Notice that, if a safe path exists, Theorem 1 guarantees that GOOSE finds the shortest one eventually.

**Synthetic data** Similarly to the setting in Turchetta et al. (2016); Wachi et al. (2018) we construct a two-dimensional grid world. At every location, the agent takes one of four actions: *left*, *right*, *up* and *down*. We use the state augmentation in Turchetta et al. (2016) to define a constraint over state

transitions. The constraint function is a sample from a GP with mean $\mu = 0.6$ and RBF kernel with lengthscale $l = 2$ and variance $\sigma^2 = 1$. If the agent takes an unsafe action, it ends up in a failure state, otherwise it moves to the desired adjacent state. We make the constraint independent of the direction of motion, i.e., $q(\mathbf{x}, \mathbf{x}') = q(\mathbf{x}', \mathbf{x})$. We generate 800 worlds by sampling 100 different constraints for square maps with sides of $20, 30, 40, \cdots, 90$ tiles and a source-target pair for each one.

We show the geometric mean of the performance of SEO and GOOSE relative to SMDP as a function of the world size in Fig. 5. Fig. 5b shows that GOOSE needs a factor 2.5 fewer samples than SMDP. Fig. 5c shows that the overhead to compute the heuristic of GOOSE is negligible, while the solution of the two MDPs [1] required by SEO is computationally intense. Fig. 5a shows that SEO outperforms GOOSE in terms of cost of the exploration trajectory. This is expected as SEO aims to minimize it, while GOOSE optimizes the sample-efficiency. However, it is easy to modify the heuristic of GOOSE to consider the exploration cost by, for example, reducing the priority of a state based on its distance from the current location of the agent. In conclusion, GOOSE leads to a drastic improvement in performance with respect to the previously known safe exploration strategy with exploration guarantees, SMDP. Moreover, it achieves similar or better performance than SEO while providing exploration guarantees that SEO lacks.

**Mars exploration**     We simulate the exploration of Mars with a rover. In this context, communication delays between the rover and the operator on Earth make autonomous exploration extremely important, while the high degree of uncertainty about the environment requires the agent to consider safety constraints. In our experiment, we consider the *Mars Science Laboratory* MSL (2007, Sec. 2.1.3), a rover deployed on Mars that can climb a maximum slope of $30°$. We use Digital Terrain Models of Mars available from McEwen et al. (2007).

We use a grid world similar to the one introduced above. The safety constraint is the absolute value of the steepness of the slope between two locations: given two states $\mathbf{x}$ and $\mathbf{x}'$, the constraint over the state transition is defined as $q(\mathbf{x}, \mathbf{x}') = |H(\mathbf{x}) - H(\mathbf{x}')|/d(\mathbf{x}, \mathbf{x}')$, where $H(\mathbf{x}), H(\mathbf{x}')$ indicate the altitudes at $\mathbf{x}$ and $\mathbf{x}'$ respectively and $d(\mathbf{x}, \mathbf{x}')$ is the distance between them. We set conservatively the safety constraint to $q(\mathbf{x}, \mathbf{x}') \geq -\tan^{-1}(25°)$. The step of the grid is 10m. We use square maps from 16 different locations on Mars with sides between 100 and 150 tiles. We generate 64 scenarios by sampling 4 source-target pairs for each map . We model the steepness with a GP with Matérn kernel with $\nu = 5/2$. We set the hyperprior on the lengthscale and on the standard deviation to be $Lognormal(30\text{m}, 0.25\text{m}^2)$ and $Lognormal(\tan(10°), 0.04)$, respectively. These encode our prior belief about the surface of Mars. Next, we take 1000 noisy measurements at random locations from each map, which, in reality, could come from satellite images, to find a maximum a posteriori estimator of the hyperparameters to fine tune our prior to each site.

In Tab. 1, we show the geometric mean of the performance of SEO and GOOSE relative to SMDP. The results confirm those of the synthetic experiments but with larger changes in performance with respect to the benchmark due to the increased size of the world.

## 5   Conclusion

We presented GOOSE, an add-on module that enables existing interactive machine learning algorithms to safely explore the decision space, without violating *a priori* unknown safety constraints. Our method is provably safe and learns about the safety of decisions suggested by existing, unsafe algorithms. As a result, it is more data-efficient than previous safe exploration methods in practice.

**Aknowlegment.**     This research was partially supported by the Max Planck ETH Center for Learning Systems and by the European Research Council (ERC) under the European Union's Horizon 2020 research and innovation programme grantagreement No 815943.

## Footnotes

[1]We use policy iteration. Policy evaluation is performed by solving a sparse linear system with SciPy (Virtanen et al., 2019). At iteration $t$, we initialize policy iteration with the optimal policy from $t - 1$.

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
