[Supplementary Material · neurips19-supplementary-camera_ready.pdf]

In the following we present the proof of our result.

## A  Definitions

For ease of consultation we repeat the relevant definitions here. We denote with $\mathcal{G} = (\mathcal{D}, E)$ the directed graph describing the dependency among decisions introduced in Sec. 2, where $\mathcal{D}$ indicates the vertices of the graph, i.e., the decision space of the problem and $E \subseteq \mathcal{D} \times \mathcal{D}$ denotes the edges. Baseline for safety:

$$R_\epsilon^{\text{safe}}(S) = S \cup \{\mathbf{x} \in \mathcal{D} \setminus S, | \exists \mathbf{z} \in S \colon q(\mathbf{z}) - \epsilon - Ld(\mathbf{x}, \mathbf{z}) \geq 0\}, \tag{3}$$

The ergodicity operator is defined by intersecting the nodes that are reachable from a set $S$ and the nodes from which we can return to a set $\overline{S}$ through a path contained in another set $S$:

$$R^{\text{reach}}(S) = S \cup \{\mathbf{x} \in \mathcal{D} \setminus S \,|\, \exists \mathbf{z} \in S : (\mathbf{z}, \mathbf{x}) \in E\}, \tag{4}$$

$$R_n^{\text{reach}}(S) = R^{\text{reach}}(R_{n-1}^{\text{reach}}(S)) \text{ with } R_1^{\text{reach}}(S) = R^{\text{reach}}(S), \tag{5}$$

$$\tilde{R}^{\text{reach}}(S) = \lim_{n \to \infty} R_n^{\text{reach}}(S), \tag{6}$$

$$R^{\text{ret}}(S, \overline{S}) = \overline{S} \cup \{\mathbf{x} \in S \,|\, \exists \mathbf{z} \in \overline{S} : (\mathbf{x}, \mathbf{z}) \in E\} \tag{7}$$

$$R_n^{\text{ret}}(S, \overline{S}) = R^{\text{ret}}(S, R_{n-1}^{\text{ret}}(S, \overline{S})), \text{ with } R_1^{\text{ret}}(S, \overline{S}) = R^{\text{ret}}(S, \overline{S}), \tag{8}$$

$$\tilde{R}^{\text{ret}}(S, \overline{S}) = \lim_{n \to \infty} R_n^{\text{ret}}(S, \overline{S}), \tag{9}$$

$$R^{\text{ergodic}}(S, \overline{S}) = \tilde{R}^{\text{reach}}(\overline{S}) \cap \tilde{R}^{\text{ret}}(S, \overline{S}). \tag{10}$$

Here, we repeat the definition of the safe and ergodic baseline introduced by Turchetta et al. (2016):

$$R_\epsilon(S) = R_\epsilon^{\text{safe}}(S) \cap R^{\text{ergodic}}(R_\epsilon^{\text{safe}}(S), S), \tag{11}$$

$$R_\epsilon^n(S) = R_\epsilon(R_\epsilon^{n-1}(S)) \text{ with } R_\epsilon^1 = R_\epsilon(S), \tag{12}$$

$$\overline{R}_\epsilon(S) = \lim_{n \to \infty} R_\epsilon^n(S). \tag{13}$$

Optimistic and pessimistic constraint satisfaction operators:

$$o_t^\epsilon(S) = \{\mathbf{x} \in \mathcal{D}, | \exists \mathbf{z} \in S : u_t(\mathbf{z}) - Ld(\mathbf{x}, \mathbf{z}) - \epsilon \geq 0\}, \tag{14}$$

$$p_t^\epsilon(S) = \{\mathbf{x} \in \mathcal{D}, | \exists \mathbf{z} \in S : l_t(\mathbf{z}) - Ld(\mathbf{x}, \mathbf{z}) - \epsilon \geq 0\} \tag{15}$$

Optimistic expansion operator:

$$O_t^{\epsilon,1}(S) = o_t^\epsilon(S) \cap R^{\text{ergodic}}(o_t^\epsilon(S), S), \tag{16}$$

$$O_t^{\epsilon,n}(S) = o_t^\epsilon(O_t^{\epsilon,n-1}(S)) \cap R^{\text{ergodic}}(o_t^\epsilon(O_t^{\epsilon,n-1}(S)), S), \tag{17}$$

$$\tilde{O}_t^\epsilon(S) = \lim_{n \to \infty} O_t^{n,\epsilon}(S) \tag{18}$$

Pessimistic expansion operators:

$$P_t^{\epsilon,1}(S) = p_t^\epsilon(S) \cap R^{\text{ergodic}}(p_t^\epsilon(S), S), \tag{19}$$

$$P_t^{\epsilon,n}(S) = p_t^\epsilon(P_t^{\epsilon,n-1}(S)) \cap R^{\text{ergodic}}(p_t^\epsilon(P_t^{\epsilon,n-1}(S)), S), \tag{20}$$

$$\tilde{P}_t^\epsilon(S) = \lim_{n \to \infty} P_t^{n,\epsilon}(S) \tag{21}$$

Pessimistic and optimistic safe and ergodic sets:

$$\bar{S}_t^{o,\epsilon} = \tilde{O}_t^\epsilon(\bar{S}_{t-1}^p) \tag{22}$$

$$\bar{S}_t^p = \tilde{P}_t^0(\bar{S}_{t-1}^p), \tag{23}$$

Points with uncertainty above threshold:

$$W_t^\epsilon = \{\mathbf{x} \in \bar{S}_t^p : w_t(\mathbf{x}) > \epsilon\}. \tag{24}$$

Set of decisions with equal priority:

$$A_t(\alpha) = \{\mathbf{x} \in \bar{S}_t^{o,\epsilon} \setminus p_t^0(, \bar{S}_t^p) : h_t(\mathbf{x}) = \alpha\} \tag{25}$$

Immediate expanders for nodes with priority $\alpha$:

$$G_t^\epsilon(\alpha) = \{\mathbf{x} \in W_t^\epsilon, \,|\, \exists\, \mathbf{z} \in A_t(\alpha) \colon u_t(\mathbf{x}) - Ld(\mathbf{x}, \mathbf{z}) \geq 0\}., \tag{26}$$

Relevant priority:

$$\alpha^* = \max\ \alpha, \ \text{s.t.} \ |G_t^\epsilon(\alpha)| > 0. \tag{27}$$

Notice that our definition of $\bar{S}_t^p$ differs slightly from the one in Turchetta et al. (2016) in that we alternate the pessimistic expansion step and the restriction to ergodic nodes until convergence, whereas Turchetta et al. (2016) do it only once at each time step. In particular, we have $\bar{S}_t^p = \lim_{n \to \infty} P_t^{0,n}(\bar{S}_{t-1}^p)$, while they use $\bar{S}_t^p = P_t^{0,1}(\bar{S}_{t-1}^p)$. In practice, this does not make any difference since it is easy to verify that, by $t^*$, i.e., the time by when both GoOSE and the approach in Turchetta et al. (2016) are guaranteed to converge in the worst case, the pessimistic ergodic safe sets should be the same for both methods. However, our new definition allows for a more efficient exploration. These new definitions would require us to show again some of the properties that were shown by Turchetta et al. (2016) for $\bar{S}^p$. However, due to our recursive definition of $\bar{S}^p$, it is easy to see that it is possible to show these properties by induction over the index $n$. In this case, the lemmas introduced by Turchetta et al. (2016) constitute the base case. At this point, it is sufficient to use the induction hypothesis and the monotonicity of the confidence interval shown in Lemma 1 together with basic properties of the $R^{\text{ergodic}}$ operator discussed in Lemma 4 to prove the induction step. We show how to do this explicitly in Lemma 5. However, we do not explicitly repeat this reasoning for every lemma involving $\bar{S}^p$ and we refer to Turchetta et al. (2016) instead.

## B  Preliminary lemmas

This section contains some basic lemmas about the sets defined above that will be used in subsequent sections to prove our main results.

**Lemma 1.** $\forall \boldsymbol{x} \in \mathcal{D}$, $u_{t+1}(\boldsymbol{x}) \leq u_t(\boldsymbol{x})$, $l_{t+1}(\boldsymbol{x}) \geq l_t(\boldsymbol{x})$, $w_{t+1}(\boldsymbol{x}) \leq w_t(\boldsymbol{x})$.

*Proof.* See Lemma 1 in Turchetta et al. (2016). □

**Lemma 2.** *Given* $S \subseteq R \subseteq \mathcal{D}$ *and* $\overline{S} \subseteq \overline{R} \subseteq \mathcal{D}$, *it holds that* $\tilde{R}^{\text{ret}}(\overline{S}, S) \subseteq \tilde{R}^{\text{ret}}(\overline{R}, R)$.

*Proof.* See Lemma 7 in Turchetta et al. (2016) □

**Lemma 3.** *For any* $S, R \subseteq \mathcal{D}$, *for any* $n \geq 1$ *we have* $R_n^{\text{reach}}(S) \subseteq R_n^{\text{reach}}(R)$.

We proceed by induction. For $n = 1$, we have $R^{\text{reach}}(S) \subseteq R^{\text{reach}}(R)$ by Lemma 8 by Turchetta et al. (2016). For the inductive step, assume $R_{n-1}^{\text{reach}}(S) \subseteq R_{n-1}^{\text{reach}}(R)$. Consider $\mathbf{x} \in R_n^{\text{reach}}(S)$. We know $\exists \mathbf{x}' \in R_{n-1}^{\text{reach}}(S) \subseteq R_{n-1}^{\text{reach}}(R)$, $a \in \mathcal{A}(\mathbf{x}')$ such that $\mathbf{x} = f(\mathbf{x}', a)$, which implies $\mathbf{x} \in R_n^{\text{reach}}(R)$.

**Corollary 2.** *For any* $S, R \subseteq \mathcal{D}$, *we have* $\tilde{R}^{\text{reach}}(S) \subseteq \tilde{R}^{\text{reach}}(R)$.

**Lemma 4.** *For any* $S, R \subseteq \mathcal{D}$ *and* $\overline{S}, \overline{R} \subseteq \mathcal{D}$, *we have* $R^{ergodic}(S, \overline{S}) \subseteq R^{ergodic}(R, \overline{R})$.

*Proof.* This follows from Lemma 2 and Corollary 2. □

**Lemma 5.** *For any* $t \geq 1$, $\bar{S}_0^p \subseteq \bar{S}_t^p \subseteq \bar{S}_{t+1}^p$.

*Proof.* Lemma 9 in Turchetta et al. (2016) allows us to say $P_t^{0,1}(\bar{S}_{t-1}^p) \subseteq P_{t+1}^{0,1}(\bar{S}_t^p)$. Thus, we can assume $P_t^{0,n-1}(\bar{S}_{t-1}^p) \subseteq P_{t+1}^{0,n-1}(\bar{S}_t^p)$ as induction hypothesis. Let us consider $\mathbf{x} \in P_t^{0,n-1}(\bar{S}_{t-1}^p)$. We know there exists $\mathbf{z} \in P_t^{0,n-1}(\bar{S}_{t-1}^p) \subseteq P_{t+1}^{0,n-1}(\bar{S}_t^p)$ such that $l_t(\mathbf{z}) - Ld(\mathbf{x}, \mathbf{z}) \geq 0$, which, by Lemma 1, implies $l_{t+1}(\mathbf{z}) - Ld(\mathbf{x}, \mathbf{z}) \geq 0$. This means that $p_t^0(P_t^{0,n-1}(\bar{S}_{t-1}^p)) \subseteq p_{t+1}^0(P_{t+1}^{0,n-1}(\bar{S}_t^p))$. Applying Lemma 4, we complete the induction step and show $P_t^{0,n}(\bar{S}_{t-1}^p) \subseteq P_{t+1}^{0,n}(\bar{S}_t^p)$. □

**Lemma 6.** *(Chowdhury and Gopalan, 2017, Thm. 2) Assume* $\|q\|_k^2 \leq B_q$, *and* $\sigma$-*sub-Gaussian noise. If* $\beta_t^{1/2} = B_q + 4\sigma\sqrt{\gamma_t + 1 + \ln(1/\delta)}$, *then, for all* $t > 0$ *and all* $\boldsymbol{x} \in \mathcal{D}$, $|q(\boldsymbol{x}) - \mu_{t-1}(\boldsymbol{x})| \leq \beta_t^{1/2}\sigma_{t-1}(\boldsymbol{x})$ *holds with probability at least* $1 - \delta$.

*Proof.* See Theorem 2 in Chowdhury and Gopalan (2017). □

**Lemma 7.** *Let $\beta_t^{1/2} = B_q + 4\sigma\sqrt{\gamma_t + 1 + \ln(1/\delta)}$ and assume $\|q\|_k^2 \leq B_q$, and $\sigma$-sub-Gaussian noise. Then, for all $t > 0$ and all $\boldsymbol{x} \in \mathcal{D}$, it holds with probability at least $1 - \delta$ that $q(\boldsymbol{x}) \in C_t(\boldsymbol{x})$.*

*Proof.* See Corollary 1 in Sui et al. (2015). □

# C  Safety

The safety of our algorithm depends on the confidence intervals and on the safe and ergodic set $\bar{S}_t^p$. Since these are defined as in Turchetta et al. (2016), their safety guarantees carry over to our case.

**Theorem 2.** *For any node $\boldsymbol{x}$ along any trajectory induced by Alg. 1 on the graph $\mathcal{G}$ we have, with probability at least $1 - \delta$, that $q(\boldsymbol{x}) \geq 0$.*

*Proof.* See Theorem 2 in Turchetta et al. (2016). □

# D  Completeness

In this section, we develop the core of our theoretical contribution. The analysis in Turchetta et al. (2016) bounds the uncertainty of the expanders when the safe set does not change in an interval $[t_0, t_1]$ without considering the measurements collected prior to $t_0$. By considering this information, we extend their worst case sample complexity bound to our more general formulation of the safe exploration problem.

The following lemmas refer to the exploration steps, i.e., when the goal suggested by the oracle $\mathcal{O}$ lies outside of the pessimistic safe and ergodic set (Line 5, Alg. 1). Notice that $t$ denotes the number of constraint evaluations and it differs from the iteration index of the algorithm $i$.

The core idea is the following: We bound the number constraint evaluations required at point in the domain to guarantee that its uncertainty is below $\epsilon$. We show that, as a consequence, if the safe and ergodic set does not change for long enough all the expanders have uncertainty below $\epsilon$. At this point we can either guarantee that the safe set expands or that the whole $\tilde{R}_\epsilon(S_0)$ has been explored. Since the analysis relies on the number of constraint evaluations at each point in the domain, we can evaluate them in any order as long as we exclude those that have an uncertainty below $\epsilon$. Therefore, our exploration guarantees hold for any priority function.

In the following, let us denote with $\mathcal{T}_t^{\mathbf{x}} = \{\tau_1, \cdots, \tau_j\}$ the set of steps where the constraint $q$ is evaluated at $\mathbf{x}$ by step $t$. Moreover, we assume, without loss of generality, $k(\mathbf{x}, \mathbf{x}) \leq 1$, i.e., we assume bounded variance.

**Lemma 8.** *For any $t \geq 1$ and for any $\boldsymbol{x} \in \mathcal{D}$, it holds that $w_t(\boldsymbol{x}) \leq \sqrt{\frac{C_1 \gamma_t \beta_t}{|\mathcal{T}_t^{\boldsymbol{x}}|}}$, with $C_1 = 8/\log(1 - \sigma^{-2})$.*

*Proof.*

$$|\mathcal{T}_t^{\mathbf{x}}|w_t^2(\mathbf{x}) \leq \sum_{\tau \in \mathcal{T}_t^{\mathbf{x}}} w_\tau^2(\mathbf{x}) \tag{28}$$

$$\leq \sum_{\tau \in \mathcal{T}_t^{\mathbf{x}}} 4\beta_\tau \sigma_{\tau-1}^2(\mathbf{x}), \tag{29}$$

$$\leq \sum_{\tau \leq t} 4\beta_\tau \sigma_{\tau-1}^2(\mathbf{x}), \tag{30}$$

$$\leq C_1 \gamma_t \beta_t, \tag{31}$$

with $C_1 = 8/\log(1 - \sigma^{-2})$. Here, (28) holds because of Lemma 1, and (31) holds because of Lemma 5.4 by Srinivas et al. (2010). □

For the remainder of the paper, let us denote with $T_t$ the smallest positive integer such that $\frac{T_t}{\beta_{t+T_t}\gamma_{t+T_t}} \geq \frac{C_1}{\epsilon^2}$, with $C_1 = 8/\log(1-\sigma^{-2})$ and with $t^*$ the smallest positive integer such that $t^* \geq |\tilde{R}_0(S_0)|T_{t^*}$.

**Lemma 9.** *For any $t \leq t^*$, for any $\boldsymbol{x} \in \mathcal{D}$ such that $|\mathcal{T}_t^{\boldsymbol{x}}| \geq T_{t^*}$, it holds that $w_t(\boldsymbol{x}) \leq \epsilon$.*

*Proof.* Since $T_t$ is an increasing function of $t$ (Sui et al., 2015), we have $|\mathcal{T}_t^{\boldsymbol{x}}| \geq T_{t^*} \geq T_t$. Therefore, using Lemma 8 and the definition of $T_t$, we have

$$w_t(\mathbf{x}) \leq \sqrt{\frac{C_1\gamma_t\beta_t}{T_t}} \leq \sqrt{\frac{C_1\gamma_t\beta_t\epsilon^2}{C_1\gamma_{t+T_t}\gamma_{t+T_t}}} \leq \sqrt{\frac{\gamma_t\beta_t}{\gamma_{t+T_t}\gamma_{t+T_t}}} \, \epsilon \leq \epsilon, \tag{32}$$

where the last inequality comes from the fact that both $\beta_t$ and $\gamma_t$ are increasing functions of $t$. $\square$

**Lemma 10.** *For any $t \leq t^*$, $|\mathcal{T}_t^{\boldsymbol{x}}| \leq T_{t^*}$, for any $\boldsymbol{x} \in \bar{S}_t^p$.*

*Proof.* According to Line 6 of Alg. 2, we only evaluate the constraint at points $\mathbf{x} \in \mathcal{D}$ if $w_t(\mathbf{x}) > \epsilon$. From Lemma 9 we know that $|\mathcal{T}_t^{\mathbf{x}}| = T_{t^*} \implies w_t(\mathbf{x}) \leq \epsilon$. Thus, if $|\mathcal{T}_t^{\mathbf{x}}| = T_{t^*}$, $\mathbf{x}$ is not evaluated anymore, which means that $|\mathcal{T}_t^{\mathbf{x}}|$ cannot grow any further. $\square$

**Lemma 11.** $\forall t \geq 0$, $\bar{S}_t^p \subseteq \overline{R}_0(S_0)$ *with probability at least $1-\delta$.*

*Proof.* See Lemma 22 in Turchetta et al. (2016). $\square$

The following lemma bounds the uncertainty of the points sampled by GOOSE when the set of safe and ergodic points does not increase.

**Lemma 12.** *For any $t \leq t^*$, let $\tau_t = |\bar{S}_t^p|T_{t^*}$, if $\bar{S}_t^p = \bar{S}_{\tau_t}^p$, then $w_{\tau_t}(\boldsymbol{x}) \leq \epsilon$ for all $\boldsymbol{x} \in \cup_\alpha G_{\tau_t}^\epsilon(\alpha)$.*

*Proof.* First, we notice that

$$\sum_{\mathbf{x} \in \bar{S}_{\tau_t}^p} |\mathcal{T}_{\tau_t}^{\mathbf{x}}| = \tau_t = |\bar{S}_t^p|T_{t^*} = |\bar{S}_{\tau_t}^p|T_{t^*}, \tag{33}$$

where the first equality comes from the fact that the sum of the number of constraint observations in $\tau_t$ time steps is equal to $\tau_t$, the second comes from the definition of $\tau_t$ and the third comes from the assumption that $\bar{S}_t^p = \bar{S}_{\tau_t}^p$. This allows us to say that, for all $\mathbf{x} \in \bar{S}_{\tau_t}^p$

$$\sum_{\mathbf{z} \in \bar{S}_{\tau_t}^p \setminus \{\mathbf{x}\}} |\mathcal{T}_{\tau_t}^{\mathbf{z}}| = |\bar{S}_{\tau_t}^p|T_{t^*} - |\mathcal{T}_{\tau_t}^{\mathbf{x}}|. \tag{34}$$

Moreover, we have $\tau_t = |\bar{S}_t^p|T_{t^*} \leq |\tilde{R}_0(S_0)|T_{t^*} \leq t^*$ by definition of $t^*$ and $\tau_t$ and Lemma 11. Therefore, by Lemma 10 we know that $T_{t^*} \geq |\mathcal{T}_{\tau_t}^{\mathbf{x}}|$ for all $\mathbf{x} \in \bar{S}_{\tau_t}^p$. Now we show by contradiction that $T_{t^*} = |\mathcal{T}_{\tau_t}^{\mathbf{x}}|$ for all $\mathbf{x} \in \bar{S}_{\tau_t}^p$. Assume this is not the case, i.e., there is $\mathbf{x} \in \bar{S}_{\tau_t}^p$ such that $T_{t^*} > |\mathcal{T}_{\tau_t}^{\mathbf{x}}|$. We have

$$(|\bar{S}_{\tau_t}^p| - 1)T_{t^*} \geq \sum_{\mathbf{z} \in \bar{S}_{\tau_t}^p \setminus \{\mathbf{x}\}} |\mathcal{T}_{\tau_t}^{\mathbf{z}}| = |\bar{S}_{\tau_t}^p|T_{t^*} - |\mathcal{T}_{\tau_t}^{\mathbf{x}}| > |\bar{S}_{\tau_t}^p|T_{t^*} - T_{t^*} = (|\bar{S}_{\tau_t}^p| - 1)T_{t^*}, \tag{35}$$

which is a contradiction and proves our claim that $T_{t^*} = |\mathcal{T}_{\tau_t}^{\mathbf{x}}|$ for all $\mathbf{x} \in \bar{S}_{\tau_t}^p$. Therefore, by Lemma 9, $w_{\tau_t}(\mathbf{x}) \leq \epsilon$ for all $\mathbf{x} \in \bar{S}_{\tau_t}^p$. This proves our claim since $\cup_\alpha G_{\tau_t}^\epsilon(\alpha) \subseteq \bar{S}_{\tau_t}^p$. $\square$

**Lemma 13.** *For any $t \geq 1$, $\overline{R}_\epsilon(S_0) \setminus \bar{S}_t^p \neq \emptyset$, then, $R_\epsilon(\bar{S}_t^p) \setminus \bar{S}_t^p \neq \emptyset$.*

*Proof.* See Lemma 20 in Turchetta et al. (2016). $\square$

**Lemma 14.** *For any $t \leq t^*$, if $\tilde{R}_\epsilon(S_0) \setminus \bar{S}_t^p \neq \emptyset$, then $\bar{S}_t^p \subset \bar{S}_{|\hat{S}_t|T_{t^*}}^p$ with probability at least $1-\delta$.*

*Proof.* This proof is analogous to Lemma 21 in Turchetta et al. (2016) where we use our Lemma 12 rather than their Lemma 19 to bound the uncertainty of the expanders. $\square$

**Lemma 15.** *There exists $t \leq t^*$ such that $\tilde{R}_\epsilon(S_0) \subseteq \bar{S}_t^p$ with probability at least $1 - \delta$.*

*Proof.* For the sake of contradiction, assume this is not the case and that $\forall t \leq t^*$ holds that $\tilde{R}_\epsilon(S_0) \setminus \bar{S}_t^p \neq \emptyset$. For all $i \geq 1$ define $\tau_i = |\bar{S}_{\tau_{i-1}}^p| T_{t^*}$ with $\tau_0 = 0$. We know that $\tau_i \leq t^*$ for all $i$ because of Lemma 11 and that $\tau_0 \leq \tau_1 \leq \cdots$ because of Lemma 5. Therefore, Lemma 14 implies that $\bar{S}_0^p \subset \bar{S}_{\tau_1}^p \subset \bar{S}_{\tau_2}^p \subset \cdots$. In general, this means that $|\bar{S}_{\tau_i}^p| \geq |\bar{S}_0^p| + i$ for all $i \geq 1$. In particular if we set $i = |\tilde{R}_0(S_0) \setminus S_0| + 1$ we get that $|\bar{S}_{\tau_i}^p| \geq |\bar{S}_0^p| + |\tilde{R}_0(S_0) \setminus S_0| + 1 = |\tilde{R}_0(S_0)| + 1 > |\tilde{R}_0(S_0)|$. This is a contradiction because of Lemma 11. $\square$

**Lemma 16.** *There is $t \leq t^*$ such that $\tilde{R}_\epsilon(S_0) \subseteq \bar{S}_t^p \subseteq \overline{R}_0(S_0)$ with probability at least $1 - \delta$.*

*Proof.* The lemma follows directly from Lemmas 11 and 15. $\square$

**Lemma 17.** $\bar{S}_{t^*}^{o,\epsilon} \subseteq \bar{S}_{t^*}^p$.

*Proof.* Since the optimistic and ergodic and safe set is defined recursively, we will prove this claim by induction. Similarly to Lemma 12, we start by noticing that, for every $\mathbf{x} \in \bar{S}_{t^*}^p$, we have:

$$\sum_{\mathbf{x} \in \bar{S}_{t^*}^p} |\mathcal{T}_{t^*}^{\mathbf{x}}| = t^* \geq |\tilde{R}_0(S_0)| T_{t^*}, \tag{36}$$

$$\implies \sum_{\mathbf{z} \in \bar{S}_{t^*}^p \setminus \mathbf{x}} |\mathcal{T}_{t^*}^{\mathbf{z}}| \geq |\tilde{R}_0(S_0)| T_{t^*} - |\mathcal{T}_{t^*}^{\mathbf{x}}| \geq |\bar{S}_{t^*}^p| T_{t^*} - |\mathcal{T}_{t^*}^{\mathbf{x}}|. \tag{37}$$

Lemma 10 allows us to say $|\mathcal{T}_{t^*}^{\mathbf{x}}| \leq T_{t^*}$ for all $\mathbf{x} \in \bar{S}_{t^*}^p$. We show by contradiction that $|\mathcal{T}_{t^*}^{\mathbf{x}}| = T_{t^*}$ for all $\mathbf{x} \in \bar{S}_{t^*}^p$. Assume this is not the case and that we have $\mathbf{x} \in \bar{S}_{t^*}^p$ such that $|\mathcal{T}_{t^*}^{\mathbf{x}}| < T_{t^*}$:

$$(|\bar{S}_{t^*}^p| - 1) T_{t^*} \geq \sum_{\mathbf{z} \in \bar{S}_{t^*}^p \setminus \{\mathbf{x}\}} |\mathcal{T}_{t^*}^{\mathbf{z}}| \geq |\bar{S}_{t^*}^p| T_{t^*} - |\mathcal{T}_{t^*}^{\mathbf{x}}| > |\bar{S}_{t^*}^p| T_{t^*} - T_{t^*} = (|\bar{S}_{t^*}^p| - 1) T_{t^*}, \tag{38}$$

which is a contradiction and proves that $|\mathcal{T}_{t^*}^{\mathbf{x}}| = T_{t^*}$ for all $\mathbf{x} \in \bar{S}_{t^*}^p$. Therefore, by Lemma 9, we know that $w_{t^*}(\mathbf{x}) \leq \epsilon$ for any $\mathbf{x} \in \bar{S}_{t^*}^p$. Now consider $\mathbf{x} \in o_{t^*}^\epsilon(\bar{S}_{t^*-1}^p)$. We know that there is $\mathbf{z} \in \bar{S}_{t^*-1}^p \subseteq \bar{S}_{t^*}^p$ such that $u_{t^*}(\mathbf{z}) - Ld(\mathbf{x}, \mathbf{z}) - \epsilon \geq 0$. Since $w_{t^*}(\mathbf{z}) \leq \epsilon$, we know that $l_{t^*}(\mathbf{z}) - Ld(\mathbf{x}, \mathbf{z}) \geq 0$, i.e., $\mathbf{x} \in p_{t^*}^0(\bar{S}_{t^*-1}^p)$. Using Lemma 4, we can say $O_{t^*}^{\epsilon,1}(\bar{S}_{t^*-1}^p) \subseteq P_{t^*}^{0,1}(\bar{S}_{t^*-1}^p)$. Now, we can make the following induction hypothesis: $O_{t^*}^{\epsilon,n-1}(\bar{S}_{t^*-1}^p) \subseteq P_{t^*}^{0,n-1}(\bar{S}_{t^*-1}^p)$. Consider $\mathbf{x} \in O_{t^*}^\epsilon(\bar{S}_{t^*-1}^p)$, we know there is $\mathbf{z} \in O_{t^*}^{\epsilon,n-1}(\bar{S}_{t^*-1}^p) \subseteq P_{t^*}^{0,n-1}(\bar{S}_{t^*-1}^p) \subseteq \bar{S}_{t^*}^p$ (where the first inclusion comes from the induction hypothesis and the second by definition of the safe set), such that $u_{t^*}(\mathbf{z}) - Ld(\mathbf{x}, \mathbf{z}) - \epsilon \geq 0$. Since $w_{t^*}(\mathbf{z}) \leq \epsilon$, we know that $l_{t^*}(\mathbf{z}) - Ld(\mathbf{x}, \mathbf{z}) \geq 0$, i.e., $\mathbf{x} \in p_{t^*}^0(P_{t^*}^{0,n-1}(\bar{S}_{t^*-1}^p))$. We can apply Lemma 4 again to complete the induction step and show $O_{t^*}^{\epsilon,n}(\bar{S}_{t^*-1}^p) \subseteq P_{t^*}^{0,n}(\bar{S}_{t^*-1}^p)$ and, therefore, $\bar{S}_{t^*}^{o,\epsilon} \subseteq \bar{S}_{t^*}^p$. $\square$

**Lemma 18.** *For every $t \geq 0$, we have $\tilde{R}_\epsilon(S_0) \subseteq \bar{S}_t^{o,\epsilon}$.*

*Proof.* We will show this claim with a proof by induction. Let us consider $\mathbf{x} \in R_\epsilon(S_0)$. We know there is a $\mathbf{z} \in S_0$ such that $q(\mathbf{z}) - Ld(\mathbf{x}, \mathbf{z}) - \epsilon \geq 0$. By Lemma 7, we know this means $u_t(\mathbf{z}) - Ld(\mathbf{x}, \mathbf{z}) - \epsilon \geq 0$ for all $t \geq 0$. Therefore, $R_\epsilon^{\mathrm{safe}}(S_0) \subseteq o_t^\epsilon(S_0) \subseteq o_t^\epsilon(\bar{S}_{t-1}^p)$ since $S_0 \subseteq \bar{S}_{t-1}^p$ for all $t \geq 1$ by Lemma 5. Lemma 4 allows us to say that $R_\epsilon(S_0) \subseteq O_t^\epsilon(\bar{S}_{t-1}^p)$. As induction hypothesis, we can assume $R_\epsilon^{n-1}(S_0) \subseteq O_t^{\epsilon,n-1}(\bar{S}_{t-1}^p)$. Consider $\mathbf{x} \in R_\epsilon^n(S_0)$. We know there is $\mathbf{z} \in R_\epsilon^{n-1}(S_0) \subseteq O_t^{\epsilon,n-1}(\bar{S}_{t-1}^p)$ such that $q(\mathbf{z}) - Ld(\mathbf{x}, \mathbf{z}) - \epsilon \geq 0$ which, by Lemma 7, means that $u_t(\mathbf{z}) - Ld(\mathbf{x}, \mathbf{z}) - \epsilon \geq 0$ for all $t \geq 0$. Therefore, $R_\epsilon^{\mathrm{safe}}(R_\epsilon^{n-1}(S_0)) \subseteq o_t^\epsilon(O_t^{\epsilon,n-1}(\bar{S}_{t-1}^p))$. Using Lemma 4, we can say $R_\epsilon^n(S_0) \subseteq O_t^{\epsilon,n}(\bar{S}_{t-1}^p)$, which completes the induction step and concludes the proof. $\square$

# E    Main result

**Theorem 1.** *Assume that $q(\cdot)$ is L-Lipschitz continuous w.r.t. $d(\cdot,\cdot)$ with $\|q\|_k \leq B_q$, $\sigma$-sub-Gaussian noise, $S_0 \neq \emptyset$, $q(\boldsymbol{x}) \geq 0$ for all $\boldsymbol{x} \in S_0$, and that, for any two decisions $\boldsymbol{x},\boldsymbol{x}' \in S_0$, there is a path in the graph $\mathcal{G}$ connecting them within $S_0$. Let $\beta_t^{1/2} = B_q + 4\sigma\sqrt{\gamma_t + 1 + \ln(1/\delta)}$, then, for any $h_t : \mathcal{D} \to \mathbb{R}$, with probability at least $1 - \delta$, we have $q(\boldsymbol{x}) \geq 0$ for any $\boldsymbol{x}$ visited by* GOOSE. *Moreover, let $t^*$ be the smallest integer such that $\frac{t^*}{\beta_{t^*}\gamma_{t^*}} \geq \frac{C\,|\tilde{R}_0(S_0)|}{\epsilon^2}$, with $C = 8/\log(1 + \sigma^{-2})$, then there exists a $t \leq t^*$ such that, with probability at least $1 - \delta$, $\tilde{R}_\epsilon(S_0) \subseteq \bar{S}_t^{o,\epsilon} \subseteq \bar{S}_t^p \subseteq \tilde{R}_0(S_0)$.*

*Proof.* The safety is a direct consequence of Theorem 2. The convergence of the pessimistic and optimistic approximation of the safe sets is a direct consequence of Lemmas 16–18.    □

The following corollary gives a simpler interpretation of our main result: in presence of an unknown constraint, the IML oracle augmented with GOOSE behaves as the IML oracle would behave if it had knowledge of a better-than-$\epsilon$-accurate approximation of the safe reachable set from the beginning (except for a finite number of constraint evaluations).

**Corollary 1.** *Under the assumptions of Theorem 1, let the IML oracle be deterministic given the observations. Then there exists a set $S$ with $\tilde{R}_\epsilon(S_0) \subseteq S \subseteq \tilde{R}_0(S_0)$ so that $\boldsymbol{x}_i^\star = \mathcal{O}(S)$ for all $k \geq 1$.*

*Proof.* This is a direct consequence of Theorem 1, since it guarantees that, if necessary, we can expand the set of points where we can evaluate the objective (i.e. the pessimistic safe set) and we can contract the decision space of the IML oracle (i.e. the optimistic safe set) to a point where the first contains the second, in a finite number of constraint evaluations.    □