[Reviews · NeurIPS 2019]

Reviewer 1



Originality: While safe interactive machine learning has been investigated before, I really like the reductionist approach taken here which uses any unsafe IML algorithm as an oracle and combines it with safety testing to make provably safe decisions. This reductionist approach is original. Adequate citation of earlier works has been provided. Quality: The submission is technically sound. Clarity: Most parts of the paper are well written. I really like the detailed explanation of the algorithm. However, it is still a pretty hard paper to read. The authors use a lot of ideas from a previously published paper Turchetta (2016) and unless the reader is very well aware with previous techniques it is hard to follow some of the arguments. For example, the safe expansion strategy is hard to follow w/o prior knowledge. Lines 184-203 are hard to understand w/o prior knowledge. Similarly, the choice of heuristic (lines 225- 236) paragraph was also hard to understand. Furthermore, it is not clear to me about the computational aspects of the proposed algorithm. For example, what does the safe expansion steps entail computationally? How easy is it to maintain and update the sets S_t^o and S_t^p? Significance: This is a significant result in the area of safe interactive learning.

Reviewer 2



Overall, the paper is tackling an important problem. Safe exploration is particularly important since it allows autonomous agents to gather data about their environment without violating safety constraints. The paper’s theoretical guarantees are also strong, ensuring that it finds the optimal solution within a slightly constrained version of the safe set. The assumptions appear to be standard. My main concern with this paper is its comparison to prior work, in particular, (Berkenkamp et al., 2017). The authors state that (Berkenkamp et al., 2017) “can benefit from more efficient, goal-oriented exploration”. Is this difference reflected in the theoretical guarantees provided by the authors? If not, don’t the authors at least empirically compare to this prior work to show that they obtain better solutions more quickly? I would be much more positive about this paper is the authors could clearly convey the difference between their paper and this prior work.

Reviewer 3



This paper considers the safe exploration problem in both (Bayesian, Gaussian Process) optimization and reinforcement learning settings. In this work, as with some previous works, which states are safe is treated as unknown, but it is assumed that safety is determined by a sufficiently smooth constraint function, so that evaluating (exploring) a point may be adequate to ensure that nearby points are also safe on account of smoothness. Perhaps the most significant aspect of this work is the way the problem is formulated. Some previous works allowed unsafe exploration, provided that a near-optimal safe point could be identified; other works treated safe exploration as the sole objective, with finding the optimal point within the safe region as an afterthought. The former model is inappropriate for many reinforcement learning applications in which the learning may happen on-line in a live robotic platform and safety must be ensured during execution; the latter model is simply inefficient, which is in a sense the focus of the evaluation in this work. The approach here is to take an unsafe optimization algorithm, together with a heuristic function for the cost of reaching a state (in the sense of heuristic search, eg., the A* algorithm) and use this to produce an optimization algorithm that is guaranteed to remain in safe states during the process. They use GPs over both the constraint function to guarantee safety, but also over the potential payoff (plus exploration cost) to prioritize queries. The paper provides theoretical guarantees that the search process remains safe and bounds the complexity of exploring the state space, similar to the guarantees provided by prior works. What is different is that empirically, the new method uses much fewer evaluations and much less computation than the prior works by Sui et al. for GPs; fewer evaluations than the work by Turchetta et al. for MDPs that provided such guarantees; and less computation and samples to first path (but slightly worse exploration cost) than a prior algorithm by Wachi et al. that did not feature a theoretical guarantee for exploration. In the last case, the differences in performance correspond to the respective objectives sought by the different works. In any case, the proposed method appears to be general (it can be employed with a variety of different unsafe optimization algorithms) and effective in practice. There are some small issues with the exposition. First, there seems to be some confusion between the notation used in the exposition versus in the formal theorem statement: the theorems use \bar{R} whereas the text uses \tilde{R}. Also the parameter gamma is used without really being introduced; it seems that perhaps given a value of beta a value of gamma can be inferred via the formula on line 211, I wonder how I should interpret it. These don't affect the main message of the paper too seriously though.

[Author Response · NeurIPS 2019]

We thank the reviewers for their helpful comments which we will address in our final submission, as outlined below.

**Reviewer 1:**   We are going to improve the expansion and heuristic sections in the final submission.

A naive implementation of the expansion step can be computationally expensive, since we have to compute the potential
expanders for multiple values of $p$ until we find one such that $|g_t(W_t^\epsilon, A_t(p))| > 0$. In our implementation, we keep
track of the nodes such that $|g_t(W_t^\epsilon, A_t(p))| = 0$ to make this step efficient. Further improvements are possible by
exploiting properties of the kernel (or metric $d(\cdot, \cdot)$), which often encodes that expandability is a local property. Similar
locality considerations can greatly improve the efficiency in many parts of the algorithm.

We keep track of $\bar{S}_t^o$ and $\bar{S}_t^p$ using the graph library `networkx` and by adding/removing edges as we acquire new data
points. Since at each iteration only few edges are removed/added, this step is not computationally expensive. We are
going to add details of an efficient implementation to the appendix and will release our code.

**Reviewer 2:**   We agree that there are similarities between GOOSE and the method by Berkenkamp 2017 (B17), in
that we use the same statistical analysis tools to build accurate confidence intervals and guarantee reachability-based
safety constraints during exploration. However, there are fundamental differences between the two works.

The most important difference is that of exploration. While B17 provides safety guarantees in the continuous domain,
for the exploration analysis they discretize the domain. Thus, in our context, their method can be thought of as an
extension of SAFEOPT by Sui 2015 and SAFEMDP by Turchetta 2016 to the continuous domain. While the three
methods focus on safety in different settings and thus use different tools to construct pessimistic estimates of the safe
set, they all collect data within their respective safe set estimates by following the same strategy that provably explores
the entire safe set. In particular, they all expand the safe set by reducing the maximal uncertainty within the current
safe set[1], which is easy to analyze but can be very data inefficient in practice. Thus, B17 addresses the problem of
safe identification of dynamical systems with continuous state-action spaces by learning about the transition dynamics
uniformly over the domain. This is evident from their exploration guarantees in Theorem 4 iii), which hold exclusively
for the exploration strategy introduced in equation (6) and which is the same as in SAFEOPT and SAFEMDP.

In contrast, GOOSE is a safety add-on layer with strong safety and completeness guarantees that focuses on improving
the sample efficiency in exploration. In particular, our analysis allows *any* goal-directed exploration strategy to be used
in order to efficiently learn about the part of the domain that is relevant toward the achievement of a given goal. Since
the method B17 fundamentally builds on a discrete exploration analysis, it might be possible to use the more efficient
exploration scheme of GOOSE in their setting too. While this would require more analysis due to inherent challenges
in continuous domains, GOOSE provides a first step in this direction. Thus, we think that our work is complementary
to B17 rather than overlapping with it.

There are several other differences between the two approaches: *(i)* Different sources of uncertainty and, therefore, of
risk. GOOSE considers safety-critical external environments with known transition dynamics, while B17 addresses
uncertainty in the dynamics but does not account for external factors. *(ii)* Different constraints: B17 focuses on stability
constraints, a specific type of constraint that is relevant in dynamical systems. GOOSE looks at level sets of generic
unknown functions, which can model safety constraint in a variety of IML scenarios that may not involve dynamical
systems, including BO and active learning. *(iii)* Different assumptions: GOOSE assumes a known model and, under this
assumption, stability constraints have been extensively studied. B17 assumes that a Lyapunov function, whose choice
implicitly affects the quality of the estimate of the asymptotically stable region, is given. Moreover, B17 implicitly
assumes that state action pairs are safely reachable, which is not required for GOOSE.

**Reviewer 3:**   $\gamma_t$ denotes the information capacity of the safety constraint, which is an information theoretic complexity
measure of the encoded function class. We use it quantify how many data points we require to learn the function up to a
certain accuracy. We will clarify this in the final submission.

Without additional assumptions on the environment, it is not possible to provide a bound that does not require complete
exploration. As a counter example, consider a graph $\mathcal{G}$ that is a chain of nodes, where each node only provides
information about the safety of the next one in the chain. Thus, if the unsafe IML oracle suggests a node at the end of
the chain, we must individually learn about each node in the chain to expand the safe set, which is fundamentally what
the bound in Theorem 1 encodes. In the general case, this is intuitive if one thinks of SMDP by Turchetta 2016 as the
safe equivalent of breadth first search and GOOSE as the equivalent of $A^*$. Depending on the graph, the location of the
target and the heuristic, both breadth first search and $A^*$ may need to visit all the nodes in the graph to find a path to the
target. Similarly, in our worst case analysis, it may be necessary for both GOOSE and SMDP to learn about the safety
of all nodes in the graph before evaluating the oracle suggestion.

## Footnotes

[1]SAFEOPT additionally considers maximizers and expanders for efficiency, but their analysis focuses on complete exploration.


[Meta-Review · NeurIPS 2019]

The reviewer all agree that this is solid work, and should definitely be accepted.